# Nanoparticle Tracking Analysis of Urinary Extracellular Vesicle Proteins as a New Challenge in Laboratory Medicine

**DOI:** 10.3390/ijms241512228

**Published:** 2023-07-31

**Authors:** Kornelia Sałaga-Zaleska, Agnieszka Kuchta, Beata Bzoma, Gabriela Chyła-Danił, Anna Safianowska, Agata Płoska, Leszek Kalinowski, Alicja Dębska-Ślizień, Maciej Jankowski

**Affiliations:** 1Department of Clinical Chemistry, Medical University of Gdansk, Debinki Street 7, 80-211 Gdansk, Poland; kornelia.salaga-zaleska@gumed.edu.pl (K.S.-Z.);; 2Department of Nephrology, Transplantology and Internal Diseases, Medical University of Gdansk, Smoluchowskiego Street 17, 80-214 Gdansk, Poland; 3Clinical of Nephrology, Transplantology and Internal Diseases, University Clinical Centre in Gdansk, Smoluchowskiego Street 17, 80-214 Gdansk, Poland; 4Department of Medical Laboratory Diagnostic—Fahrenheit Biobank BBMRI.pl, Medical University of Gdansk, Debinki Street 7, 80-211 Gdansk, Poland; 5BioTechMed Centre, Department of Mechanics of Materials and Structures, Gdansk University of Technology, Narutowicza Street 11/12, 80-233 Gdansk, Poland

**Keywords:** urinary extracellular vesicle, uEVs, nanoparticle-tracking analysis, renal dysfunction, chronic kidney disease

## Abstract

Urinary extracellular vesicle (uEV) proteins may be used as specific markers of kidney damage in various pathophysiological conditions. The nanoparticle-tracking analysis (NTA) appears to be the most useful method for the analysis of uEVs due to its ability to analyze particles below 300 nm. The NTA method has been used to measure the size and concentration of uEVs and also allows for a deeper analysis of uEVs based on their protein composition using fluorescence measurements. However, despite much interest in the clinical application of uEVs, their analysis using the NTA method is poorly described and requires meticulous sample preparation, experimental adjustment of instrument settings, and above all, an understanding of the limitations of the method. In the present work, we demonstrate the usefulness of an NTA. We also present problems encountered during analysis with possible solutions: the choice of sample dilution, the method of the presentation and comparison of results, photobleaching, and the adjustment of instrument settings for a specific analysis. We show that the NTA method appears to be a promising method for the determination of uEVs. However, it is important to be aware of potential problems that may affect the results.

## 1. Introduction

Extracellular vesicles (EVs) are a highly heterogeneous group of membrane-bound particles that can be broadly divided into several categories:-apoptotic bodies: originating from the plasma membrane; 50–5000 nm in diameter;-ectosomes: pinched off from the surface of the plasma membrane by outward budding; 50–1000 nm in diameter; subdivided into microvesicles, microparticles, and large vesicles;-exosomes: endosomal origin; 30–100 nm in diameter.

Vesicles are released by cells into all body fluids, such as urine, blood, saliva, and amniotic fluid [1,2]. The vesicles secreted into the urine, called urinary extracellular vesicles (uEVs), come from various parts of the urinary system, particularly the kidneys [3].

Extracellular vesicles are involved in immune signaling, angiogenesis, the removal of unwanted or damaged substances, and proliferation by stimulating cells to secrete biologically active substances, such as cytokines or growth factors, thus ensuring intercellular communication and the maintenance of cellular homeostasis [4,5]. During EV formation, vesicles incorporate various bioactive molecules from their cell of origin, including the membrane, cytosolic and nuclear proteins, nucleic acids (DNA, mRNA, and noncoding RNA), lipids (e.g., lysophosphatidylcholine), and metabolites [6]. As numerous studies in recent years have shown, analyses of the amount of proteins in uEVs may be useful as noninvasive, specific markers in various pathophysiological conditions. Markers that allow the identification of isolated structures as uEVs and their quantitative expression use the tetraspanin surface family. Proteomic analyses have indicated that uEVs are mainly of a urogenital origin [5,7,8]. In addition, most uEVs come from the kidneys, not from the urinary tract, as shown by comparing the number of vesicles in urine obtained with a nephrostomy drain to that obtained from the whole urinary tract [3]. It appears to be important in the search for early markers of, for example, podocyte dysfunction, where a loss of more than 30% leads to impaired permeability of the glomerular filtration barrier to proteins and cannot be compensated for with differentiating progenitor cells [9,10,11]. Therefore, uEV-based markers are currently being investigated for a number of renal diseases or, especially, diseases that affect renal function. Proteomic analyses of uEVs have demonstrated their clinical significance:-Wilms tumor protein in diabetic nephropathy [12], glomerulonephritis, exacerbation of focal segmental glomerulosclerosis, and steroid-sensitive nephrotic syndrome [13];-cystatin B and NGAL in type-1 diabetes [14];-regucalcin [15], alpha-1-microglobulin/bikunin precursor (AMBP), MLL3, and VDAC1 proteins [16] in type-1 and type-2 diabetes;-aminopeptidase and vasopressin precursor in thin basement membrane disease [17];-ceruloplasmin and α-1-antitrypsin in IgA nephropathy [17];-aquaporin-2 forms in IgA nephropathy [18];-fetuin A in acute renal failure [19];-CD 133 in end-stage renal disease and after kidney transplantation [20,21].

The methods used to analyze the proteins of uEVs are flow cytometry, Western blot analysis, dynamic light scattering (DLS), and nanoparticle-tracking analysis (NTA) [6,22]. Flow cytometry, which is based on the analysis of laser light scanning a particle, appears ideal for the analysis of EVs; however, it is routinely suitable for larger particles of over 300 nm, making it unsuitable for the analysis of uEVs, where the smallest EVs are of greatest interest [23]. Western blot analysis provides a semiquantitative assessment of uEV proteins but does not determine their diameter or the absolute amount in a given sample. The DLS method, in which particle size is determined through fluctuations in the scattered light intensity, is very sensitive to the polydispersity of a sample—the presence of larger particles or aggregates makes it impossible to determine the size of much smaller particles, such as uEVs [24]. The most useful method for the analysis of uEVs appears to be an NTA, which is dedicated to the analysis of extracellular vesicles below 300 nm. The NTA was first commercialized in 2006 [24] and is characterized by very high accuracy in determining the size and concentration of both monodisperse and polydisperse samples [24].

Recent publications have highlighted the wide potential diagnostic and therapeutic applications of uEVs and the NTA as a method for their analysis. A study by van der Pol et al. based on the calibration of an instrument with polystyrene beads showed that, in polydisperse samples, the concentration of smaller vesicles was underestimated, while the concentration of larger vesicles was overestimated. In this system, accurate quantification of the number of smallest vesicles was also a problem. Studies of analogous samples of polystyrene beads in polydisperse systems using other available methods have yielded divergent results for vesicle concentrations due to differences in their minimum detectable size. Therefore, although flow cytometry provides the most accurate measurements, allowing the detection of 270–600 nm vesicles with conventional flow cytometry and 150–190 nm with a special small particle attachment of limited availability, the NTA method, which detects 70–90 nm vesicles, is applicable to clinical uEV studies [25].

Analyses carried out on polystyrene beads with different camera settings have shown that the results of an NTA analysis depend on the settings used and that the software offers the possibility of presenting the data in a summary file using mathematical operations that affect the interpretation of the results, e.g., rolling average or finite track length adjustment (FTLA—tracking a particle over a finite number of frames) [25,26]. The applied solutions have improved the visual reading of the data by smoothing and narrowing the peaks in particle concentration versus size plots. Maguire et al. highlighted the effects of temperature, viscosity of the solvent, and diffusion coefficient in obtaining reproducible NTA size measurements, which are essential for the comparability of particle sizes over time and space, by plotting these variables on an Ishikawa cause-and-effect analysis graph. They also emphasized that the introduction of an FTLA algorithm, knowledge of the basics of the method, and obtaining the results could facilitate repeatable measurements for an NTA [24]. Therefore, it is important to emphasize the need for the operator to understand the measurement characteristics of an NTA to avoid misinterpretation of the results, especially when comparing data. Based on tests performed in different laboratories, Vestad et al. emphasized that the standardization of instrument settings and their selection according to sample characteristics is essential to achieve satisfactory repeatability, reproducibility, and reliable determination of particle concentration and that the differences obtained may be due to the subjective choice of measurement settings [27]. Similar observations were made when colorectal and glioblastoma cell lines were tested to alter EV secretion and the use of an NTA alone was found to be insufficient [28]. Jan Lötvall et al. pointed to the need to develop a standard for EVs that would allow the comparison of EV analyses from different laboratories. One limitation to their development was the lack of universal and unambiguously specific EV markers, which depend on the material from which they are isolated. The authors indicated that they are likely to be developed in the future, as knowledge of the composition of EVs improves. They also pointed out that, at present, results should be reported together with technical information on camera settings in order to maintain the reproducibility of analyses [29].

An NTA simultaneously provides information on the number or concentration of particles, their size, the polydispersity of a sample, the fluorescence signal, and the relative light intensity, so it seems to be ideal for the analysis of nanoparticles, such as uEVs. At the same time, proteins of uEVs could be useful as noninvasive, specific markers for the early detection of kidney injury in various pathophysiological conditions. The aim of this manuscript is to provide evidence that the NTA is a useful method for the analysis of uEVs and to highlight the potential problems that may lead to unreliable results.

## 2. Results

Data on the expressions of the specific proteins CD 63, CD 9, and podocin in the uEVs are presented in Figure 1. A Western blot analysis of all the investigated samples using anti-CD 63, CD 9, and anti-podocin antibodies demonstrated positive signals.

Figure 2 shows an example of a sample analysis result using a nanoparticle-tracking analysis. The average size of the uEVs was approximately 126 nm (86–245 nm). The total number of uEVs was counted in a sample diluted to 1:100 with 75 ± 3 particles per reading frame and was 5.2 × 10^11^ ± 1.9 × 10^10^ particles/mL.

The effect of dilution on the number of particles per frame counted with the nanoparticle-tracking analysis software is shown in Figure 3. Depending on the dilution of the sample, the average numbers of particles in the reading frame were 42, 55, and 199 for dilutions of 1:1000, 1:500, and 1:100, respectively. The total measurements of the number of particles per milliliter were similar for the 1:500 and 1:1000 dilutions, with a difference of less than 5%. The final result of the total number of particles per milliliter in the 1:100 dilution was almost three times lower than those in the 1:500 and 1:1000 dilutions, with a high coefficient of variation (15% vs. 3% vs. 6%, respectively).

This study provided evidence for a positive correlation between the total number of uEVs per 24 h and creatinine-standardized uEVs (R = 0.9530, *p* = 0.0009) but not between the total number of uEVs per milliliter (R = 0.6916, *p* = 0.0852), as shown in Figure 4.

The results of the expression of the uEV-specific marker CD 63 obtained with the NTA are presented in Figure 5. Analysis of the CD 63 expression data showed that the average size of uEVs using a 488 nm laser was approximately 66 nm (17–101 nm) without a 500 nm filter (Figure 5a) and 66 nm (52–98 nm) with a 500 nm filter (Figure 5b). In addition, the total number of uEVs was 2.9 × 10^10^ ± 2.0 × 10^10^ particles/mL without a 500 nm filter (Figure 5a) and 1.5 × 10^11^ ± 1.8 × 10^9^ particles/mL with a 500 nm filter (Figure 5b). A comparison of the sizes and concentrations of uEVs without and with a 500 nm long-pass filter is shown in Figure 5c.

The results for podocin expression obtained with the NTA are presented in Figure 6. Determination of the expression of podocin uEVs, a marker of podocyte origin, in this sample showed that the average size using a 488 nm laser was approximately 156 nm (116–241 nm) without a 500 nm filter (Figure 6a) and 56 nm (39–79 nm) with a 500 nm filter (Figure 6b). The total number of uEVs was 3.5 × 10^10^ ± 3.5 × 10^9^ particles/mL without a 500 nm filter (Figure 6a) and 4.3 × 10^10^ ± 4.3 × 10^9^ particles/mL with a 500 nm filter (Figure 6b). A comparison of the sizes and concentrations of uEVs without and with a 500 nm long-pass filter is shown in Figure 6c.

## 3. Discussion

Urine as a material for obtaining extracellular vesicles would be advantageous for use in the diagnosis of kidney function. Urine is usually obtained in a noninvasive manner, and it is a material that is relatively cheap to protect, with no medical personnel being required. However, the downside of urine is the need for a sample preparation procedure that allows for maintaining the stability of uEV proteins. The classic way to protect a sample is to add a freshly prepared protease inhibitor and preservative [8,30,31]. In addition, it should be noted that analyses of signals obtained using the Western blot method have indicated a higher efficiency of uEV isolation from urine stored at −80 °C compared to urine stored at 4 °C or −20 °C, which require less available freezer space [8,32]. An additional problem in the isolation of urine samples is the formation of a protein structure of vesicle-crosslinking fibers, for example, from Tamm–Horsfall protein (uromodulin), which can trap uEVs and lead to a reduction in isolation yield. The addition of dithiothreitol (DTT) has been reported as a compound that reduces disulfide bonds, preventing the formation of uromodulin aggregates [2,30,33,34]. It is proposed to add DTT to a pellet obtained after the first centrifugation of urine, and then to incubate and centrifuge it again.

Research has described the ability to isolate EVs using an ultracentrifugation-based method [35], size-exclusion chromatography [36,37,38], polymer-based capture [35,39], and an immuno-affinity capture technique [40]. These methods are troublesome to use to isolate EVs from large sample volumes, such as urine, and the collected vesicles are usually impure [40]. Most urine samples need to be concentrated before isolation, so the most commonly chosen method for uEV isolation is the ultracentrifugation-based method [36].

The effectiveness of uEV isolation can be verified by checking the expressions of specific markers. The most commonly used markers of uEVs belong to the tetraspanin surface family: CD 9, CD 63, and CD 81 [41,42,43]. The choice of tetraspanin as a marker should depend on the origin of the uEVs analyzed. As indicated by a study by Park et al., the highest signal from human urine was obtained for tetraspanin CD 63, and it was definitely higher than those for CD 81 and CD 9 [42]. Our results confirm these previous observations, and we obtained a positive signal using Western blot for human samples with anti-CD 63 antibody (Figure 1a), as well as for the expression of CD 9 in rat samples (Figure 1b). A general overview of anti-podocin in uEVs obtained using Western blot analysis suggested their renal origin (Figure 1c).

A nanoparticle-tracking analysis is based on the visualization of light scattered from nanoparticles moving with Brownian motion and located in a laser light beam through the simultaneous use of laser light-scattering microscopy and a charge-coupled device camera. The method is based on the speed of movement of particles, which is closely related to their size, according to the Stokes–Einstein equation:$$D=\frac{4K_{B}T}{3\pi \eta d},$$
where D is the diffusion coefficient, K_B_ is the Boltzmann constant, T is temperature, η is viscosity, and d is the particle diameter [23].

This method allows the determination of the mean diameter of particles in the range of 10 to 1000 nm at a sample concentration of approximately 10^6^–10^12^ particles/mL, which is presented in Figure 2. When working with an NTA, it is important to pay attention to several crucial aspects in order to obtain reliable results. Reports from the literature, as well as our own research, give hope that analyses of the composition of uEVs with the NTA method accompanied by an understanding of the advantages and limitations of an NTA may allow targeted diagnoses of early renal dysfunction.

One of the most common problems is the adjustment of the dilution analysis based on preliminary studies, and our results showed that this was extremely important because it could affect the determination of the total number of particles per milliliter. The concentration of particles according to the manufacturer’s recommendations, should be in the range of 30–80 particles in the field of view to allow the reliable measurement of particles [44]. Too high of a concentration and, thus, too many particles in the field of view, leads to an unreliable reading due to particle overlap, which in our analysis resulted in a three-fold lower uEV concentration, as shown in Figure 3. The measuring volume of the instrument on which we carried out the measurements was 100 × 80 × 10 µm (width × length × depth), so nanoparticles with a diameter of less than 300 nm could freely overlap, especially if they were present in excess. In addition, the overlapping particles created a background that reduced the quality of the reading. Equally, however, a low concentration, meaning few molecules in the field of view, leads to an analysis based on too few observations in the set measurement time and could, therefore, also mean a measurement with a large error.

The clinical application of uEV analyses in morning urine or a random spot urine sample is definitely more convenient from the point of view of material collection than the use of a twenty-four-hour urine sample. However, reporting the results as the total number of particles per milliliter may lead to clinically misleading conclusions. The introduction of validated normalization methods is required to compare the number of vesicles, independent of urine dilution. As reported by Blijdorp et al., urinary creatinine excretion could be used to standardize the excretion of extracellular vesicle in urine samples, but it should be remembered that the use of creatinine as a factor for urine dilution could be affected by muscle mass and diet, as well as by impaired glomerular filtration rate (GFR), and this should be taken into account when selecting a study group [45]. Our study confirms the possibility of using a creatinine-standardized number of uEVs and showed a positive correlation between the total number of uEVs per 24 h and the creatinine-standardized uEVs (R = 0.9530, *p* = 0.0009) but not between the total number of uEVs per milliliter (R = 0.6916, *p* = 0.0852), as shown in Figure 4.

After analysing the size and concentration of the uEVs, the NTA allowed a deeper analysis of uEVs based on their protein composition using fluorescence measurements and allowed the counting of specifically labeled molecules. Reading in fluorescence mode is possible by using appropriate fluorophores. A fluorophore is excited by a laser and then emits energy at slightly lower frequencies and longer wavelengths. The best fluorophore has a large stoke shift, which means that the excitation and emission spectra are separated. For NTA measurement to be possible, the amplitudes of the absorption maximum and the emission maximum must be separated. Ideally, the excitation and emission spectra are completely separated. In addition, fluorophores with a high quantum yield, i.e., the ratio of the amplitude of the absorption maximum to the emission maximum, are most suitable for this work.

Fluorophores are broadly divided into organic molecules and quantum dots; in addition, the choice of laser reading depends on the fluorophore used. Organic molecules are widely available in conjugated form, but they have a tendency to photobleach, and their unbound excess can also increase background intensity. The most commonly used fluorophore for the analysis of uEV samples is Alexa Fluor 488, which is a bright green fluorescent dye with a maximum excitation wavelength of 495 nm and emission at 519 nm, making it ideal for a 488 nm laser. Other fluorophores used in the analysis of uEVs include green fluorescent protein (GFP), Rhodamines, Alexa546, and Alexa647. The second type of fluorophore is quantum dots, which are semiconductors or chalcogenides (selenides or sulphides) of metals, such as CdSe or ZnS, which must be conjugated to antibodies. They have very stable quantum yield and both unbound and bound quantum dots are measured during an NTA. A 405 nm laser is best-suited to excite quantum dots.

In a first analysis, the light scatter of a sample is measured. An analysis is then performed using a filter. Depending on the fluorophore used, a special filter can be used to measure only the longer wavelength emission from the fluorescently labeled vesicles. For example, when using an Alexa488 fluorophore, a 488 nm laser is used for excitation, and then measurements can be made without a filter, that is, measuring emissions in the entire laser wave spectrum, and with a 500 nm filter, measuring emissions at wavelengths longer than 500 nm. Each sample is usually analyzed from three to five times using the same settings. The NTA method showed uEVs to be labile, and rapid freezing at −80 °C with the addition of protease inhibitors proved to be the best approach to urine storage prior to analysis [32].

The NTA method is not without its drawbacks and one of the most common problems is photobleaching, that is, the permanent loss of a fluorophore’s ability to fluoresce due to chemical damage or photon-induced covalent modification. Overexposure to a fluorophore can cause a signal to disappear or weaken during analysis. The measurements obtained are then unique, and comparing them, e.g., in groups, may lead to erroneous conclusions. The software allows individual analyses to be recorded from among the repetitions, and it is extremely important to ensure that the number of particle counts obtained does not decrease with successive recordings. Avoiding the overexposure of a fluorophore, such as by using a synchronisation cable and a syringe pump to slow down the sample flow, can significantly reduce this problem. We performed the analysis of fluorescent samples presented in Figure 5 and Figure 6 using a syringe pump and protected the samples from overexposure.

A common problem is fluorescence background noise due to an excess of unbound fluorophores, as shown in Figure 5. The camera image visible during the analysis was not reflected in the number of counts because the contrast between the light scattered by the particles and the background was too low. An optimal signal-to-noise ratio can be achieved by experimentally selecting the appropriate ratios of primary and secondary antibodies. Too much fluorophore in a solution can increase the background intensity and drown out the signal from labeled vesicles, quench the fluorescence signal, or conversely, multiply the number of vesicles measured (Figure 5a). In our analysis, it is most likely that we obtained a count resulting from both the signal from the scattered light on the vesicles and the background noise. The solution to this problem is to label vesicles at a high concentration and dilute them immediately before measurement to achieve an optimal signal-to-noise ratio, which is shown in Figure 6b.

The results obtained indicated that the reading and analyzing of the sample delivered by the syringe pump (same syringe pump speed and dilution factor) with and without a filter using the same measurement settings (same detection threshold, slider shutter, and slider gain) resulted in divergent images (Figure 6a,b). It can be seen that, when the sample was analyzed with a 488 nm laser and a 500 nm filter, larger particles were lost from the analysis; however, small particles were numerous (Figure 6b). The same relationship appeared when the concentrations of the antibodies used and the incubation times were changed. Taking into account the influence of the settings on the results obtained, the choice of identical analysis settings seemed justified in order to eliminate the influence of the operator on the results obtained and to allow for comparison of the results obtained between samples. However, to perform measurements, it was necessary to optimize the image obtained by adjusting the sample concentration, image focus, beam position, and camera settings individually for measurements without and with the filter. The results presented in Figure 6 indicated that it was appropriate to adjust the instrument settings for the analysis of samples in scattered light (without a filter) and in fluorescence mode (with a filter). The previously discussed problem of selecting the correct sample concentration for measurement was even more important when we also wanted to analyze samples in fluorescence mode. This was because it should be taken into account that, after selecting a sample concentration for analysis without a filter, when a reading consists of all uEVs, we could expect fewer counts after applying a filter, depending on the number of uEVs expressing a given protein. Indeed, it seemed appropriate to take this dependency into account and to select the sample concentration for analysis experimentally so as not to draw conclusions about the expressions of individual proteins obtained from measurements in fluorescence mode on the basis of single counts only.

Another limitation of the NTA method, among other things, is that not only EVs, but also other particles in a sample can be counted with light scattering. Dragovic et al. showed the presence of numerous lipid vesicles, for example, chylomicrons and very-low-density lipoproteins, in human plasma, which could be detected using light scattering but not counted with fluorescence measurement [23], but were very rare in urine samples. In addition, as a proteomic analysis indicated, up to 99.96% of uEV proteins originated from the urogenital tract [7].

Thane et al. noted that biological samples, due to their polydispersity, pose a challenge to the usability of an NTA. They emphasized that, to ensure that vesicles were sufficiently illuminated with a low background share, it was crucial to select camera settings that were appropriate to the sample being analyzed. Moreover, they stressed that it was impossible to determine the same settings for all the samples. Using particle-size-related fluorescence intensity information could reflect the relative expression frequencies of surface markers of the EVs, which increased the transparency of the conducted analyses [46]. As pointed out by Bachurski et al., when conducting an NTA, one should be aware of potential errors resulting from, among others, heterogeneity of the sample and measurement settings, and in the future, NTA devices and software should be improved to ensure the standardization of measurements for biological applications while minimizing an operator’s influence on the measurements [47].

## 4. Materials and Methods

### 4.1. Ethical Approval

This study was conducted in accordance with the Declaration of Helsinki (as revised in 2013). All participants provided written informed consent to participate in the study. All study procedures were reviewed and approved by the director of University Clinical Center in Gdansk (application No. 81/2021) and the Independent Bioethics Committee for Scientific Research at the Medical University of Gdańsk (No. NKBBN/145/2022).

The animal study protocol was conducted in accordance with the European Convention for the Protection of Vertebrate Animals Used for Experimental and Other Scientific Purposes and was approved by the Ethics Committee of the local Bioethics Commission in Bydgoszcz, Poland (approval No. 44/2019, 12 December 2019).

### 4.2. Materials

Exemplary samples of rat and human urine were used to demonstrate the opportunities and challenges of a nanoparticle-tracking analysis. Human urine (first morning samples) was collected from patients at the Clinic of Nephrology, Transplantology, and Internal Diseases at the University Clinical Center in Gdańsk. Rat urine (twenty-four-hour urine sample) was collected from male Wistar rats (Tri-City Academic Laboratory Animal Center, Gdańsk, Poland) in metabolic cages (Tecniplast, Italy).

Urine was collected in tubes containing protease inhibitors (5 × 10^−4^ M PMSF, 10^−6^ M leupeptin) and preservative (3 × 10^−3^ M sodium azide) and was stored at −80 °C until used.

### 4.3. Isolation of uEVs Using Ultracentrifugation-Based Method

Urine samples were prepared using centrifugation at 1000× *g* for 15 min at 4 °C (5804 R Eppendorf Centrifuge, Hamburg, Germany) to remove mucus and epithelial cells. The samples were then centrifuged twice at 17,000× *g* for 30 min at 4 °C to remove aggregates and large bubbles above 300 nm. Proper isolation was carried out through ultracentrifugation of the samples three times at 200,000× *g* for 1 h at 4 °C (Optima TLX and MAX-XP Ultracentrifuges, Beckman Coulter, Brea, CA, USA). The uEVs were then retrieved from the pellets after gently discarding the supernatants. The pellets were suspended in PBS containing protease inhibitors.

### 4.4. Western Blot Analysis

Proteins of urinary extracellular vesicles were denatured (98 °C, 5 min), subjected to 10% sodium dodecyl sulfate-polyacrylamide gel electrophoresis, and transferred to membranes (50 min at 80 mA, TE70X Semi-dry Blotters, Hoefer Inc., Holliston, MA, USA). Next, the membranes were blocked (3% milk, 0.05% Tween-20 in TBS (Avantor Performance Material Poland S.A., Gliwice, Poland)) for 1 h at room temperature and were incubated overnight at 4 °C with rabbit primary antibodies (CD 9, SA35-08 (Invitrogen, Waltham, MA, USA) 1:500; CD 63, HPA010088, Sigma-Aldrich, Saint Louis, MO, USA, 1:1000; podocin, P0372, Sigma-Aldrich, 1:1000). After washing (0.01% Tween-20 in TBS), secondary antibodies conjugated to horseradish peroxidase (554021, BD Pharmingen (BD Biosciences, San Jose, CA, USA) 1:10,000) or alkaline phosphatase (A0545, Sigma-Aldrich, 1: 10,000) were added to the membranes for 1 h at room temperature. Reaction products were detected using a chemiluminescent substrate or an NBT/BCIP color development substrate (Thermo Fisher Scientific, Waltham, MA, USA). The membranes were analyzed and archived in a GelDoc-It Imaging System (UVP).

### 4.5. Nanoparticle-Tracking Analysis of uEVs

A NanoSight NS300 instrument (Malvern Panalytical, Malvern, UK) was used to determine the concentrations and sizes of the uEVs in the samples. The total number of extracellular vesicles was measured during the continuous flow of samples delivered from a syringe pump. Two lasers were used to visualize the light scattered on nanoparticles: 405 nm and 488 nm with or without a 500 nm long-pass filter. The samples were incubated overnight at 4 °C with rabbit primary antibodies: anti-CD 63 (HPA010088, Sigma-Aldrich) or anti-podocin (P0372, Sigma-Aldrich). Next, secondary antibodies conjugated to Alexa Fluor 488 fluorescent dye (ab150073-500, Abcam, Cambridge, MA, USA) were added to the membranes for 2 h at room temperature in the dark. The samples were diluted with PBS solution before analysis. Measurement settings (camera level, detection threshold, slider shutter, slider gain) and syringe pump speed were selected for analysis. The results obtained were the averages of five analyses with the same settings.

### 4.6. Measurement of Creatinine Concentrations

The creatinine concentration was measured using the enzymatic method (1260362 Wiener lab, Argentina), and the reading was conducted with a Multiskan GO 51119300 spectrophotometer (Thermo Fisher Scientific).

### 4.7. Statistical Analysis

Statistical analyses were performed using STATISTICA 13.3 software. Continuous variables were expressed as means ± SE (standard error) or medians and 25th and 75th percentiles. A Shapiro–Wilk test was used to test the determined normality of the distribution of variables. Correlations were assessed using standardized Pearson coefficients. Differences were considered significant for *p* < 0.05.

## 5. Conclusions

Analyses of the quantities and protein compositions of uEVs may complement the current routinely used markers of renal damage (urinary albumin excretion, blood creatinine concentration, and estimated glomerular filtration rate (eGFR)), which have become unsatisfactory in the development of therapeutic strategies. However, the low purity of the vesicles obtained and the difficulties in their analysis remain a problem. The NTA appears to be the best method currently available for the analysis of uEVs, and although measuring the expressions of individual proteins with the NTA method is not straightforward, with an understanding of the advantages and limitations of an NTA, it may become an important tool for monitoring renal function.

## Figures and Tables

**Figure 1 ijms-24-12228-f001:**
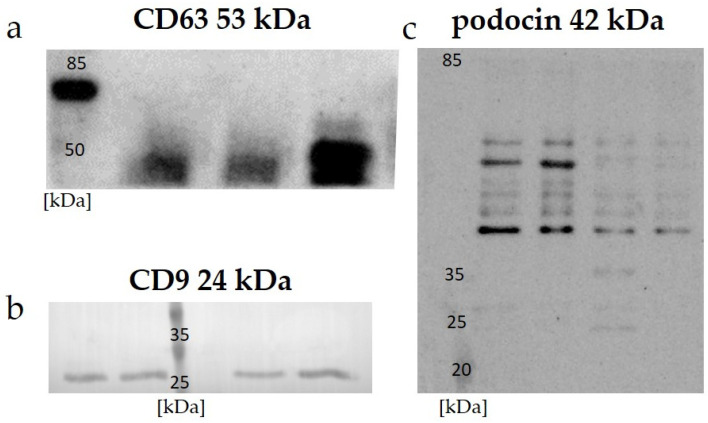
Expressions of specific markers in protein-standardized samples detected with Western blot analysis: (**a**) CD 63—examples of human uEV samples; (**b**) CD 9—examples of rat uEV samples; (**c**) podocin—examples of rat uEV samples.

**Figure 2 ijms-24-12228-f002:**
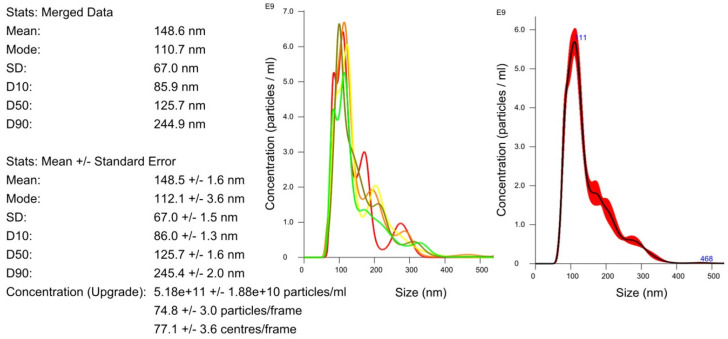
Determination of the size and concentration of uEVs: dilution factor—1:100; laser—405 nm; camera level—5; detection threshold—5; slider shutter—1206; slider gain—245; syringe pump speed—100; average of five analyses with the same settings; human uEV samples.

**Figure 3 ijms-24-12228-f003:**
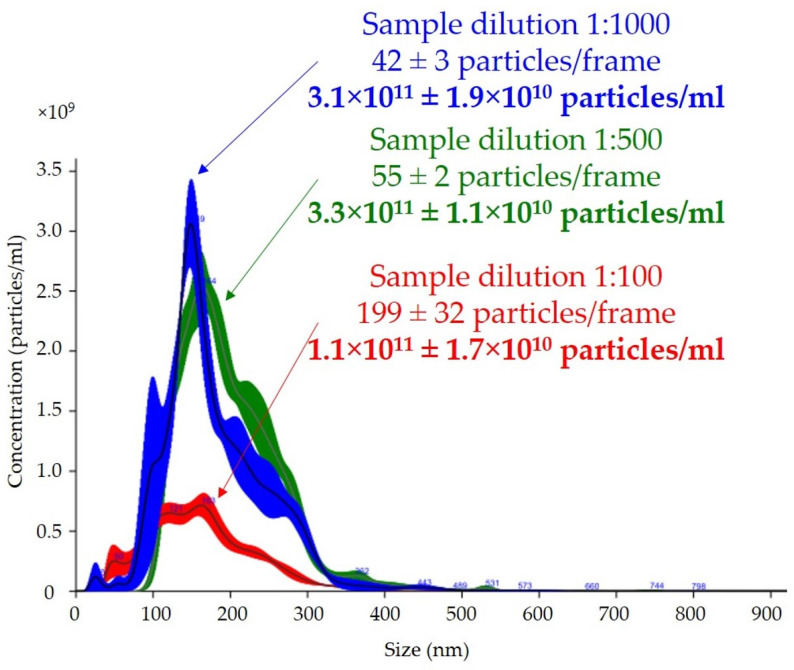
Effect of dilution on total number of particles per milliliter and size of uEVs in nanoparticle-tracking analysis: sample dilutions—1:100, 1:500, and 1:1000; laser—488 nm; camera level—14; detection threshold—5; slider shutter—1259; slider gain—245; syringe pump speed—30; average of five analyses with the same settings; human uEV samples.

**Figure 4 ijms-24-12228-f004:**
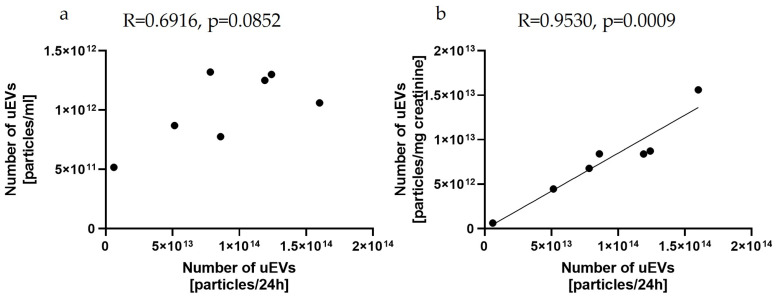
Relationships between the number of uEVs per 24 h and the number of uEVs per milliliter (**a**) and the number of uEVs per milligram of creatinine (**b**): sample dilution—1:1000; laser—488 nm; camera level—10; detection threshold—3; slider shutter—696; slider gain—55; syringe pump speed—50; average of five analyses with the same settings; rat uEV samples.

**Figure 5 ijms-24-12228-f005:**
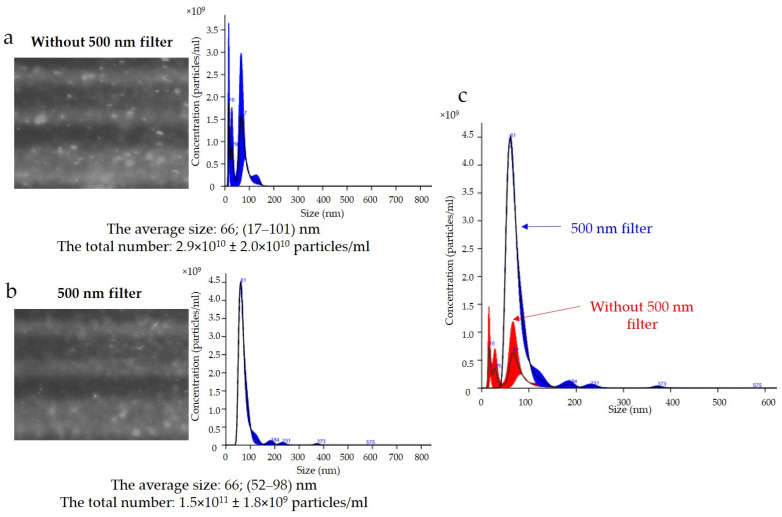
Fluorescence-based nanoparticle-tracking analysis of CD 63 expression in uEVs: (**a**) without 500 nm long-pass filter; (**b**) with 500 nm long-pass filter; (**c**) comparison of sizes and concentrations of uEVs without and with 500 nm long-pass filter; dilution factor—1:100; laser—488 nm; camera level—13; detection threshold—4; slider shutter—1232; slider gain—175; syringe pump speed—30; average of five analyses with the same settings; human uEV samples.

**Figure 6 ijms-24-12228-f006:**
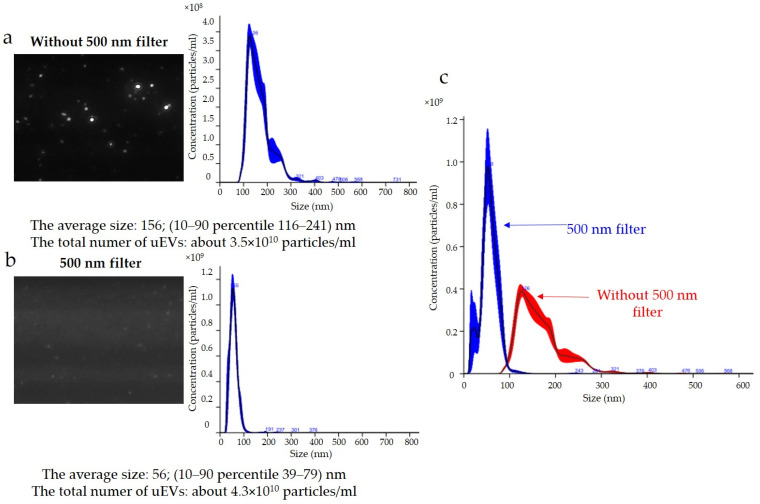
Fluorescence-based nanoparticle-tracking analysis of podocin expression in uEVs: (**a**) without 500 nm long-pass filter; (**b**) with 500 nm long-pass filter; (**c**) comparison of sizes and concentrations of uEVs without and with 500 nm long-pass filter; dilution factor—1:100; laser—488 nm; camera level—13; detection threshold—4; slider shutter—1232; slider gain—175; syringe pump speed—30; average of five analyses with the same settings; human uEV samples.

## Data Availability

The data presented in this study are available on request from the corresponding author.

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
