# Peer review of "Nanoparticle Tracking Analysis of Urinary Extracellular Vesicle Proteins as a New Challenge in Laboratory Medicine"

_ijms, 2023, doi:10.3390/ijms241512228_

Round 1

Reviewer 1 Report

The present study demonstrated the usefulness of nanoparticle tracking analysis and also presented problems encountered during the analysis with possible solutions: choice of the sample dilution, method of presentation and comparison of results, photobleaching and adjustment of instrument settings for a specific analysis. Although this MS has overall interest and visibility, some aspects should still be considered to improve the quality and objectiveness.

1)      The background of the study should be made very clear. Provide more details of the introduction and review of the work. The introduction is very poor.

2)      The figure's quality should be improved.

3)      Please speculate about the reasons for the obtained results. The discussion needs to improve.

4)      Please provide the conclusion section. In Conclusion, the authors should add the potential practical application.

5)      The article should be reviewed for English language proficiency and grammar. There are a lot of sentences without sense, misspelled words, and punctuation errors.

The present study demonstrated the usefulness of nanoparticle tracking analysis and also presented problems encountered during the analysis with possible solutions: choice of the sample dilution, method of presentation and comparison of results, photobleaching and adjustment of instrument settings for a specific analysis. Although this MS has overall interest and visibility, some aspects should still be considered to improve the quality and objectiveness.

1)      The background of the study should be made very clear. Provide more details of the introduction and review of the work. The introduction is very poor.

2)      The figure's quality should be improved.

3)      Please speculate about the reasons for the obtained results. The discussion needs to improve.

4)      Please provide the conclusion section. In Conclusion, the authors should add the potential practical application.

5)      The article should be reviewed for English language proficiency and grammar. There are a lot of sentences without sense, misspelled words, and punctuation errors.

Reviewer 2 Report

Dear Authors,

thank you for this interesting article. I believe that nanotechnology will be the future of medicine, both for diagnosis and therapy. The introduction provides enough for the chosen topic. To the Discussion section, add the name of figure - "Stokes-Einstein equation" - line 202 - 203.

There are also several articles debating the role of nanoparticle tracking analysis to assess the accuracy of urinary extracellular vesicle preparation techniques, so I think more references mean more value for your article, which appears to be seriously well done.

A conclusion section should have been individualized.

I think that article can be very useful for understanding this new tehnology.

Good luck!

Reviewer 3 Report

Considering the importance of uEV as specific renal marker for a variety of pathological conditions, your article, focused on the current possibility of analysing these proteins, represents a major step-forward in better understanding the modality of assessing and eventually applying the obtained findings in clinical practice, even if you identified some potential problems that could affect the final results. The only aspect that should be better explained is related to your group of patients - please detail the inclusion and/or exclusion criteria applied to your study participants.

Round 2

Reviewer 1 Report

Requested corrections were completed.

Requested corrections were completed.